# DataSpoon: Validation of an Instrumented Spoon for Assessment of Self-Feeding

**DOI:** 10.3390/s20072114

**Published:** 2020-04-09

**Authors:** Tal Krasovsky, Patrice L. Weiss, Oren Zuckerman, Avihay Bar, Tal Keren-Capelovitch, Jason Friedman

**Affiliations:** 1Department of Physical Therapy, University of Haifa, Haifa 3498838, Israel; 2Pediatric Rehabilitation Department, Sheba Medical Center, Tel Hashomer 52621, Israel; 3Department of Occupational Therapy, University of Haifa, Haifa 3498838, Israel; plweiss@gmail.com (P.L.W.); talkrn@gmail.com (T.K.-C.); 4Media Innovation Lab, The Interdisciplinary Center (IDC) Herzliya, Herzliya 4610101, Israel; orenz@idc.ac.il (O.Z.); avihaybar@gmail.com (A.B.); 5Efi Arazi School of Computer Science, The Interdisciplinary Center (IDC) Herzliya, Herzliya 4610101, Israel; 6Department of Physical Therapy, Sackler Faculty of Medicine, Tel Aviv University, Tel Aviv 6997801, Israel; jason@tau.ac.il

**Keywords:** kinematics, concurrent validity, outcome assessment, feasibility, rehabilitation

## Abstract

Clinically feasible assessment of self-feeding is important for adults and children with motor impairments such as stroke or cerebral palsy. However, no validated assessment tool for self-feeding kinematics exists. This work presents an initial validation of an instrumented spoon (DataSpoon) developed as an evaluation tool for self-feeding kinematics. Ten young, healthy adults (three male; age 27.2 ± 6.6 years) used DataSpoon at three movement speeds (slow, comfortable, fast) and with three different grips: “natural”, power and rotated power grip. Movement kinematics were recorded concurrently using DataSpoon and a magnetic motion capture system (trakSTAR). Eating events were automatically identified for both systems and kinematic measures were extracted from yaw, pitch and roll (YPR) data as well as from acceleration and tangential velocity profiles. Two-way, mixed model Intraclass correlation coefficients (ICC) and 95% limits of agreement (LOA) were computed to determine agreement between the systems for each kinematic variable. Most variables demonstrated fair to excellent agreement. Agreement for measures of duration, pitch and roll exceeded 0.8 (excellent agreement) for >80% of speed and grip conditions, whereas lower agreement (ICC < 0.46) was measured for tangential velocity and acceleration. A bias of 0.01–0.07 s (95% LOA [−0.54, 0.53] to [−0.63, 0.48]) was calculated for measures of duration. DataSpoon enables automatic detection of self-feeding using simple, affordable movement sensors. Using movement kinematics, variables associated with self-feeding can be identified and aid clinical reasoning for adults and children with motor impairments.

## 1. Introduction

Recent technological advances have enabled the development of lightweight, wearable inertial motion sensors, which are showing promise as rehabilitation tools [1]. Inertial sensors can monitor movement quality and present valuable information to clinicians during on-site or tele-rehabilitation sessions using affordable equipment [2].

Sensor-based assessment of movement kinematics is currently used for gait analysis in healthy individuals [3] as well as clinical populations such as Parkinson’s disease [4], stroke and Huntington’s disease [5] and children with cerebral palsy [6]. Upper limb kinematics have been recorded using wearable sensors in healthy individuals [7] as well as individuals after stroke [8], in order to objectively quantify movement patterns. Additionally, inertial sensors are able to detect performance of functional tasks such as drinking or brushing hair, in both healthy and clinical populations [9]. In future applications, information derived from low-cost inertial sensors may be able to be used to provide feedback (e.g., auditory, visual, tactile), and affect motor performance as is currently the case with more high-end motion capture systems [10].

Due to their age or physical condition, the use of body-mounted sensors is problematic for some populations who may be uncomfortable with or encumbered by the use of external measurement devices. To circumvent this problem, sensor-based technology can be embedded within everyday objects thereby creating clinically-feasible tools for the measurement of movement quality during functional movements such as eating. Existing applications of instrumented tools for eating include forks [11] and chopsticks [12] which help assess and promote fine motor skills and healthy eating habits in children.

In this study, we present the initial validation of an instrumented spoon (DataSpoon) [13,14], developed as an assessment tool for clinicians that provides quantitative information regarding self-feeding in children and adults with motor impairments such as cerebral palsy (CP) or stroke. Self-feeding is one of several self-care activities that are critical for the well-being of a child [15], hence it is an important skill to train, develop and monitor in children with motor disorders [13]. Furthermore, self-feeding kinematics is altered in people with neurological conditions, such as Parkinson’s disease [16], stroke [17], or Multiple Sclerosis [18], and in children [19] and adults [20] with cerebral palsy. Specifically, both spatial and temporal patterns of reaching with a utensil to the mouth may be altered and movements are slower, more curved and less smooth. Furthermore, due to a reduced ability to individually control the fingers, people with neurological conditions may opt for an alternative grip strategy (e.g., “power grip”) which is typical for young children [21] and leads to further changes in kinematics and force production throughout the movement [16,17,22]. Such alterations in kinematics support the need to evaluate self-feeding in people with motor impairments using a clinically-feasible measurement system. The DataSpoon system includes an instrumented spoon wirelessly paired with an Android smartphone application which presents information regarding eating patterns to a clinician. Monitoring self-feeding kinematics was demonstrated to be feasible among children of different ages and a small sample of children with CP [23]. However, before measures of self-feeding kinematics can be used to detect between-group differences in children or adults with or without motor impairments, it is essential that the psychometric properties of measuring self-feeding kinematics be established. Thus, the current work is a preliminary validation of sensor-based information from the spoon vis-à-vis a “gold standard” kinematic measurement, during self-feeding in healthy young adults. This was accomplished by: (1) describing the automated detection of feeding events from an affordable inertial sensor embedded within a teaspoon (DataSpoon) and (2) determining the validity of kinematic measures extracted from DataSpoon when compared with a “gold standard” motion capture system. We chose to evaluate kinematic measures which are associated with linear velocity and acceleration and are considered “gold standard” [24] as well as measures based on angular velocity and acceleration.

## 2. Materials and Methods

### 2.1. Participants

Ten young adults (3 male, 27.2 ± 6.6 years old) were recruited from local university students and staff. They were included in this study if they: (1) were 18–40 years of age, (2) were right-hand dominant and (3) did not have any orthopedic or neurological problems affecting arm movement kinematics or causing pain in arm movement. Ethics approval was received from the Tel-Aviv University Human ethics committee (authorization no. 11152802), and the participants signed an informed consent form before participating in the study.

### 2.2. Instruments

DataSpoon is an instrumented spoon (size: 19*2*1.5 cm, mass: 38 g; Figure 1a) which enables measurement of 6 Degrees of Freedom kinematic data and real-time presentation of movement via a smartphone app. A small, low cost wireless 3D accelerometer with gyroscope and magnetometer (red amber from GemSense, Haifa, Israel, see Table 1) is mounted at the proximal base of the spoon’s handle, and a single CR2032 replaceable battery is mounted at the distal end of the spoon’s handle. The raw 3D acceleration data and fused absolute orientation angles (in quaternions) were sampled at 50 Hz and transmitted via Bluetooth to a dedicated android smartphone and transformed to Excel files for offline processing. DataSpoon signals were compared to a “gold-standard” magnetic motion capture system (trakSTAR, Ascension Technology Corp., Shelburne, VT, USA, see Table 1). A trakSTAR sensor (Figure 1a) sampling at 200 Hz was attached to the center of the DataSpoon handle such that concurrent data was collected from both systems. The sensor’s lightweight cable was attached with medical tape to the subject’s forearm, allowing free movement of the spoon while minimizing the forces applied by the cable on the spoon.

### 2.3. Procedures

Participants were seated by a table such that both feet were flat on the floor with hips and knees flexed at 90 degrees. A plate was placed on the table such that the center of the plate aligned with the midline of the participant. A mark on the table to the right of the plate identified the initial and final position of the spoon (Figure 1a). Participants were required to eat small amounts of yoghurt/soft cheese/fruit puree using the DataSpoon at three speeds of movement (slow, comfortable, fast) and with three different grips of the spoon: “natural” grip, power grip and rotated power grip (Figure 1b–d). The power grip is a common grip used among typically developing young children [25,26] as well as children [19] and adults [16,17] with motor impairments due to neurological conditions. Due to limited range of motion in the wrist in the frontal plane (radial/ulnar deviation), this grip type allows for a smaller variety of movements [27]. The rotated power grip was intended to provide an awkward eating posture for participants in order to facilitate variable movement kinematics among healthy individuals, which may be closer to the increased variability of movement kinematics observed in people with motor impairments such as cerebral palsy [22,28]. The inclusion of varied grip positions was intended to provide a variable constraint on hand posture which may translate to variable self-feeding kinematics, and thus challenge the detection of feeding events (such as spoon in mouth) and allow for more accurate computation of validity scores. The instruction to participants was to “hold the spoon as you normally would hold a spoon” for the “natural” grip, which was typically a precision grip, to “keep the thumb below the handle and close to the spoon itself” for the power grip and to “keep the thumb below the handle and oriented towards the distal end of the spoon” for the rotated power grip. Participants performed three repetitions in each condition, such that the total number of eating cycles was ~27.

### 2.4. Data Analysis

Yaw, Pitch and Roll angles (Figure 2) were obtained directly from the trakSTAR and computed from quaternions for DataSpoon. The angles from the red amber in the DataSpoon are calculated on the device from the raw data using a proprietary algorithm. A filtered derivative for Yaw was calculated (2nd order Butterworth low-pass filter, 1 Hz cutoff) and used to detect eating events using a similar algorithm for both the trakSTAR and DataSpoon signals (Figure 3): (1) Movement onset event: the first point where both the yaw and yaw velocity signals exceed 5% of their respective peaks; (2) Spoon in mouth event: the highest peak in the yaw signal, removing adjacent peaks if inter-peak distance was under 2 s; (3) Spoon down event: time of the first zero crossing in the yaw velocity signal after each “in mouth” event. For visualization purposes, trakSTAR and DataSpoon signals were synchronized by performing a fast rotation (pitch) movement of the spoon prior to each recording. The timing of the peak in pitch was synchronized between the signals automatically using code. However, outcome measures were calculated separately for each device.

The duration of the eating phases (to- and from the mouth) and the range of pitch and roll motion were calculated from Yaw, Pitch and Roll angles (Tait–Bryan angles, Figure 2). Additional measures were extracted from the acceleration signal: in order to obtain tangential velocity, the following procedure was performed: a filtered acceleration signal (2nd order Butterworth low-pass filter, 1 Hz cutoff) was multiplied by the rotation matrix obtained from the spoon. The baseline acceleration signal was subtracted in order to eliminate the effect of gravity, and the acceleration signal was low-pass filtered (4th order Butterworth filter, 3 Hz cutoff low-pass), integrated and high-pass filtered (4th order Butterworth filter, 0.35 Hz cutoff) before calculating the square root of the sum of squares to obtain tangential velocity [29]. The peak tangential velocity was computed for each part of the movement—up (onset to in mouth) and down (in mouth to spoon down). As a measure of movement fluency (i.e., smoothness), the number of zero crossings in the acceleration profile was calculated for each movement axis and summed over the three axes (Figure 4). This number represents the number of peaks in the tangential velocity profile, which is a measure of smoothness (more peaks indicate a jerkier movement) [30].

### 2.5. Statistical Analysis

Concurrent validity was provided using two-way, mixed model Intraclass correlation coefficients (ICCs; single measures) which were computed separately for each movement condition (model ICC (3,2)) [31]; ICC values smaller than 0.4 were defined as poor, 0.41 < ICC < 0.6 as fair, 0.61 < ICC < 0.8 as good, and 0.81 < ICC < 1.0 as excellent agreements. In addition, 95% limits of agreement were calculated by averaging the measurements for each participant under each condition, subtracting the DataSpoon measurement from the trakSTAR measurement and computing mean ±1.96 standard deviations of the difference.

## 3. Results

Out of a total of 257 movements which were recorded, eight movements were unavailable due to technical issues associated with the spoon (communication lags and disconnections) and 27 movements were removed when events could not be identified reliably by either device (for example when pause between movements was too small). Thus, 222 movements were analyzed in total.

Results of ICCs are detailed in Table 2. ICCs were fair to excellent for measures of duration and range of motion, and poor to fair for measures of peak velocity and movement fluency. Mean differences and 95% Limits of Agreement are detailed in Table 3. Although the mean bias was smaller than 100 ms for temporal measures, and smaller than 1.5 degrees for angular measures (range of roll and of pitch movement in the first stage of eating), 95% limits of agreement exceeded 22 degrees for angular measures and were ~±0.5 s for temporal measures. Tangential velocity and acceleration measures showed some bias, which may have resulted from integration error for the DataSpoon.

## 4. Discussion

The current work presents an instrumented spoon which uses a simple, affordable inertial movement sensor and extracts kinematic features of movement that may be clinically important for feeding kinematics in general, and specifically for children and adults with motor disorders. The present results indicate that for most kinematic measures, concurrent validity of movement quality measures, which were extracted automatically from both systems, was fair to excellent when evaluated for young, healthy individuals. These results were similar for different grips, a natural grip and two types of power grip, designed to impose a constraint on movement kinematics which requires a modification of the motor plan. Agreement was low for measures based on tangential velocity and acceleration, as compared with yaw, pitch and roll measures. For some measures, agreement was lower for faster movements. This result merits further investigation, as it may be that for faster movements, differences between devices such as sampling rate or sensitivity may become more significant. However, one of the main differences of self-feeding patterns in people with motor impairments such as stroke [17,32] or cerebral palsy [20,22] is slowness of movement. This suggests that the larger ICCs identified for slower movements may be advantageous when evaluating self-feeding in people with motor impairments.

Recent years have witnessed the increased integration of technology into clinical practice via measures of movement quality, specifically for upper limb movement [33] and functional activities such as handwriting [34]. Affordable systems allow for objective and accurate assessment of movement quality using wearable sensors for mobility as well as upper limb movement [1]. However, a review of objective measures of upper limb functional task performance demonstrated that measurement of upper limb kinematics relies on inertial sensors in only 2.2% of the cases, whereas in 64.5% of papers published between 2002 and 2013, the instrument used was an opto-electric or magnetic motion capture system [24]. These instruments are typically expensive and require special operating conditions (e.g., somewhere to place the cameras). In order for this ratio to change, assessment of movement quality based on relatively cheap inertial sensors should rely on valid and reliable measurement. The current work demonstrated that the validity of the outcome greatly varies, specifically the validity of outcomes based on angular velocity vs. linear acceleration. Most kinematic outcomes involve measures of position (e.g., path length) or velocity (e.g., peak velocity, time to peak velocity) [24]. However, the computation of velocity and position from an inertial sensor is not a trivial problem. Inertial sensor data are characterized by drift, which accumulates when integrating acceleration to velocity and further to position. Potential solutions to this problem may include periodic recalibration of the data at rest [29] which requires manual identification of rest periods, a technique that is labor intensive. The current work takes a different approach, and shows that by using measures based on yaw, pitch and roll, better agreement can be reached between DataSpoon and a gold standard motion capture system. More work is required in order to verify whether these measures accurately capture features of self-feeding in children and adults with motor impairments. To date, we have demonstrated the initial feasibility of DataSpoon with children of different ages with and without CP [23], and future work is required to address its feasibility in other clinical populations. Indeed, in the process of designing DataSpoon, input from clinicians suggested that some of these measures (e.g., duration, smoothness) are clinically meaningful to experts in the field [13]. It should be noted, however, that the placement of the inertial sensor within the spoon itself (and not on the arm/hand complex) limits the ability to capture essential aspects of motor performance during self-feeding, such as the type of grip, or compensations related to movements of the wrist, elbow, shoulder and/or trunk [22]. This limitation is common to wearable sensors which are placed on the end-effector, but may be overcome by adding additional sensors on proximal body segments. In the current study, a single trakSTAR sensor was placed on the spoon itself in order to compare its movement with that computed by DataSpoon. Although the sensor and cable may potentially affect movement kinematics, the cable was exceptionally lightweight and the use of similar setups in many studies involving various arm movements [35,36] suggest that the effect on kinematics is minimal. An additional limitation of the current work is the somewhat heavier (38 g) weight of the DataSpoon compared with an ordinary spoon due to the added board and batteries in the handle. We expect that the effect of this weight change on external torques during self-feeding to be minimal since most of the spoon’s weight is located in the handle which is placed close to the anchor point (i.e., the hand). It is thus unlikely that the kinematics of using the DataSpoon differed significantly from that of an ordinary spoon. Furthermore, in preliminary feasibility testing with children [23] the spoon’s weight was not subjectively reported to be an issue. We therefore suggest that deviations from typical self-feeding kinematics are minimal, supporting future use of the DataSpoon by people with motor impairment.

## 5. Conclusions

Eating with a spoon is characterized by several salient kinematic features: a unidirectional change in yaw angle for each movement phase, a short-duration change in roll and in pitch angle during the initial scooping phase, followed by relatively stable roll and pitch angles during the transport to mouth phase [14]. This work shows that automatic identification of these salient events is possible. We suggest that using these measures to describe self-feeding kinematics, makes it possible to tap into functionally-relevant variables associated with efficient performance. In the future, these measures can potentially be used to provide targeted knowledge of performance feedback [32] in order to modify self-feeding performance in people with motor impairments.

In this study, measures of duration and of angular range of motion demonstrated excellent validity. Furthermore, we demonstrate here that kinematic measures based on angular velocity have higher concurrent validity compared with measures based on linear velocity and acceleration (peak velocity, fluency) when extracted from a low-cost inertial sensor, possibly due to the larger computational cost and errors associated with obtaining the latter measures. Future studies will be performed to validate this approach with atypical populations (e.g., cerebral palsy) where self-feeding kinematics are impaired [28]. In addition, additional information will be integrated to complement the DataSpoon, such as a time-coupled trunk movement sensor [22] which will assist identification of postural compensations.

## Figures and Tables

**Figure 1 sensors-20-02114-f001:**
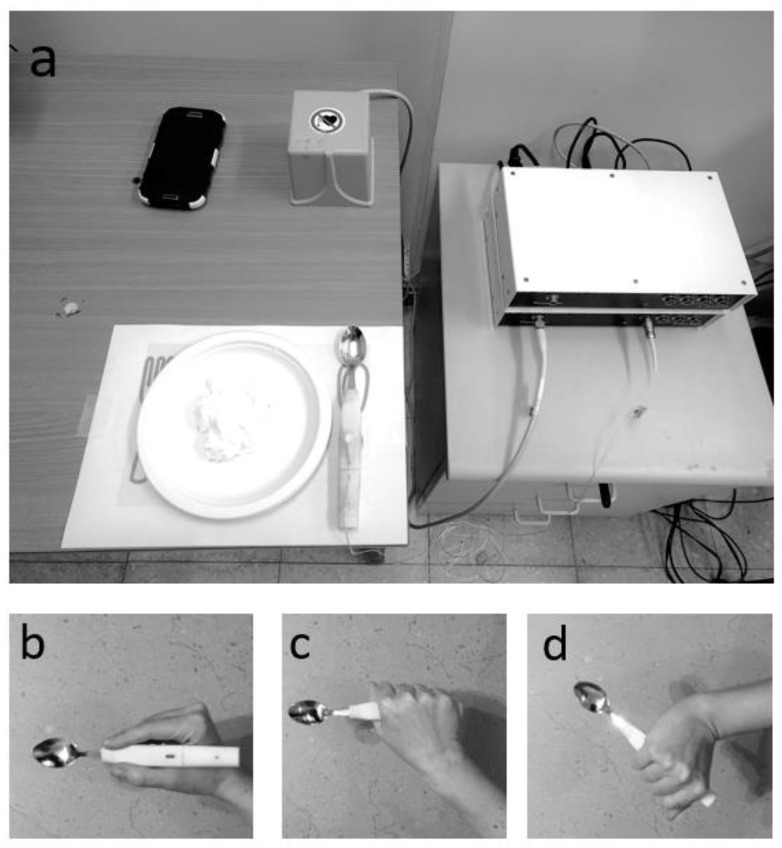
(**a**) Experimental setup. DataSpoon was placed on a placemat pointing towards the distal end of the table. A smartphone captured real-time spoon movement. A trakSTAR sensor was located at the center of the spoon and connected to the trakSTAR box via a lightweight cable. (**b**) Natural grip. (**c**) Power grip. (**d**) Rotated power grip.

**Figure 2 sensors-20-02114-f002:**
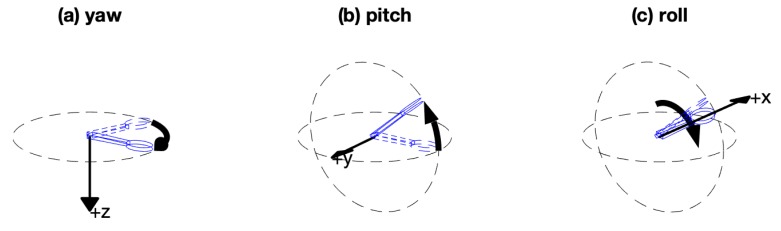
Yaw, Pitch and Roll angles (Tait–Bryan angles). The final orientation consists of three rotations in order: (**a**) yaw is the rotation about the z (up-down) axis; (**b**) pitch is the rotation about the rotated horizontal (y) axis; (**c**) roll is the rotation about the long axis (rotated x axis) of the spoon.

**Figure 3 sensors-20-02114-f003:**
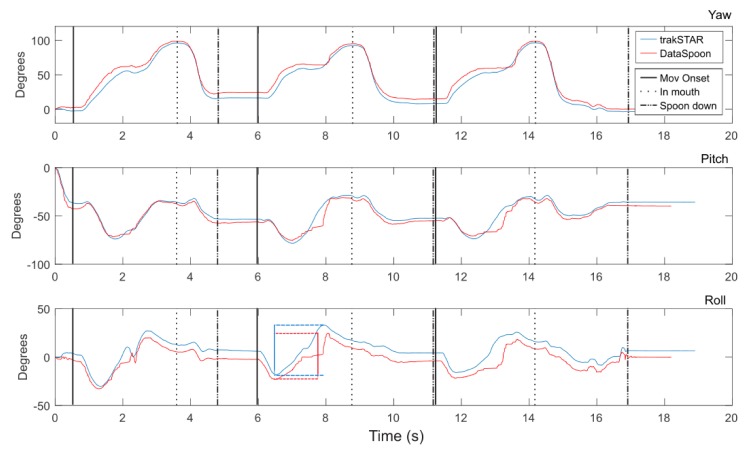
Yaw, Pitch and Roll angles for 3 consecutive eating cycles at natural spoon position and comfortable speed. trakSTAR (blue) and DataSpoon (red) signals were synchronized by a common movement of pitch at onset of recording. Black vertical lines indicate timing of eating cycle events identified for trakSTAR signals. Blue and red vertical lines (bottom panel) demonstrate the calculation of range (in this case - of roll) for one movement part.

**Figure 4 sensors-20-02114-f004:**
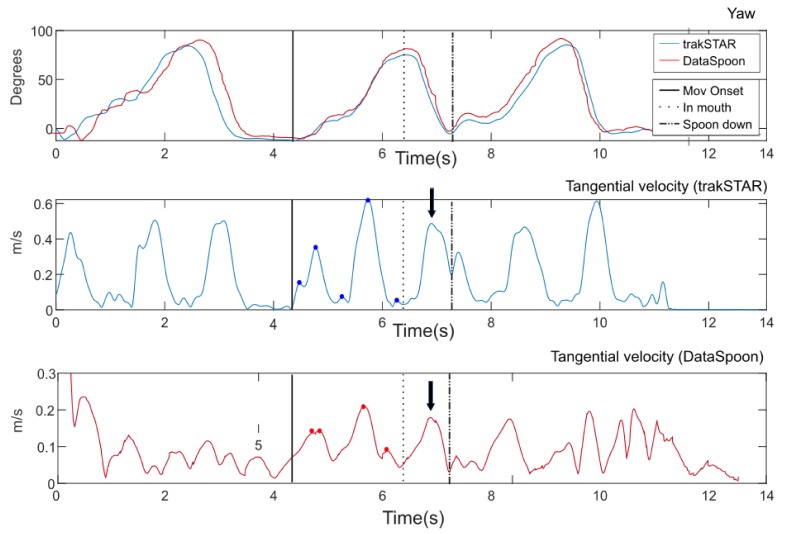
Tangential velocity profiles from trakSTAR (middle panel) and DataSpoon (bottom panel). Yaw for both systems is depicted in the top panel for comparison. One movement duration is marked for both devices. The number of peaks in the tangential velocity profile (i.e., zero crossings in the acceleration profile) is marked for the first part of movement (“to mouth”), and the peak velocity is marked for the second part (“from mouth”).

**Table 1 sensors-20-02114-t001:** Comparison of motion capture devices used in the study.

	GemSense Red Amber (Including Battery Extension)	Ascension trakSTAR System with Model 180 Sensor
Size	24 mm diameter	2 mm diameter, 9.9 mm length (not including cable)
Mass	25 g	<5 g (not including cable)
Accuracy	Not available	Position: 1.4 mm RMS, angle: 0.5° RMS
Range	Dependent on Bluetooth (approx. 10 m)	58 cm at highest accuracy level
Approximate cost	USD 40	USD 4000 (for a one-sensor setup)
Sample rate	50 Hz	200 Hz (maximum is 255 Hz)

**Table 2 sensors-20-02114-t002:** Intraclass correlation coefficients (ICCs) depicting agreement between trakSTAR and DataSpoon, with 95% confidence interval (square brackets) and significance level below. ICC values higher than 0.8 (excellent agreement) are in bold.

Measure	Natural Grip			Power Grip			Rotated Power Grip		
	Slow	Comfortable	Fast	Slow	Comfortable	Fast	Slow	Comfortable	Fast
**Duration of Movement to Mouth**	0.99 [0.99, 1.00] <0.01	0.99 [0.97, 0.99] <0.01	0.86 [0.66, 0.94] <0.01	0.95 [0.90, 0.98] <0.01	0.85 [0.66, 0.93] <0.01	0.86 [0.66, 0.94] <0.01	0.97 [0.92, 0.98] <0.01	0.98 [0.95, 0.99] <0.01	0.88 [0.77, 0.94] <0.01
**Duration of Movement from Mouth**	0.87 [0.74, 0.94] <0.01	0.83 [0.89, 0.98] <0.01	0.55 [0.16, 0.79] <0.01	0.90 [0.79, 0.95] <0.01	0.88 [0.74, 0.94] <0.01	0.50 [0.06, 0.77] 0.02	0.95 [0.89, 0.98] <0.01	0.83 [0.67, 0.92] <0.01	0.89 [0.77, 0.95] <0.01
**Duration of Movement (total)**	0.97 [0.93, 0.98] <0.01	0.99 [0.97, 0.99] <0.01	0.91 [0.80, 0.96] <0.01	0.94 [0.86, 0.97] <0.01	0.94 [0.87, 0.97] <0.01	0.87 [0.69, 0.95] <0.01	0.96 [0.89, 0.98] <0.01	0.94 [0.87, 0.97] <0.01	0.97 [0.93, 0.98] <0.01
**Range of Pitch**	0.86 [0.70, 0.93] <0.01	0.81 [0.62, 0.91] <0.01	0.62 [0.27, 0.83] <0.01	0.75 [0.52, 0.88] <0.01	0.64 [0.32, 0.83] <0.01	0.90 [0.75, 0.96] <0.01	0.87 [0.73, 0.94] <0.01	0.92 [0.83, 0.96] <0.01	0.92 [0.84, 0.96] <0.01
**Range of Roll**	0.93 [0.85, 0.97] <0.01	0.98 [0.96, 0.99] <0.01	0.50 [0.10, 0.76] 0.01	0.79 [0.58, 0.90] <0.01	0.97 [0.93, 0.99] <0.01	0.85 [0.65, 0.94] <0.01	0.95 [0.84, 0.98] <0.01	0.97 [0.94, 0.99] <0.01	0.94 [0.88, 0.97] <0.01
**Peak Velocity to Mouth**	0.24 [−0.10, 0.59] <0.01	0.21 [−0.06, 0.57] <0.01	0.07 [−0.25, 0.43] 0.35	0.21 [−0.10, 0.52] 0.04	0.05 [−0.15, 0.32] 0.33	0.37 [−0.08, 0.71] <0.01	0.07 [−0.28, 0.42] 0.36	−0.03 [−0.14, 0.16] 0.66	0.00 [−0.10, 0.16] 0.50
**Peak Velocity Down**	0.07 [−0.10, 0.30] 0.21	0.09 [−0.05, 0.35] 0.01	0.06 [−0.07, 0.29] 0.16	0.06 [−0.07, 0.26] 0.16	0.07 [−0.07, 0.30] 0.11	0.28 [−0.11, 0.65] <0.01	0.10 [−0.09, 0.36] 0.12	0.07 [−0.08, 0.30] 0.13	0.08 [−0.08, 0.30] 0.13
**Fluency - Acceleration Zero Crossing (total)**	0.14 [−0.26, 0.50] 0.25	0.41 [0.04, 0.68] <0.01	0.21 [−0.13, 0.54] 0.12	0.00 [−0.32, 0.35] 0.50	0.45 [0.07, 0.72] 0.01	0.26 [−0.21, 0.63] 0.14	−0.11 [−0.48, 0.29] 0.70	0.36 [−0.02, 0.65] <0.01	0.14 [−0.13, 0.43] 0.15

**Table 3 sensors-20-02114-t003:** Median and IQR values for the different kinematic measures for trakSTAR and DataSpoon, mean difference (bias) and 95% limits of agreement (±1.96 standard deviations) between trakSTAR and DataSpoon measurements.

Measure	Units	trakSTAR	DataSpoon	Mean Bias	95% Limits of Agreement
**Duration of Movement to Mouth**	Seconds	2.10 (0.71)	2.20 (0.70)	−0.07	[−0.51, 0.38]
**Duration of Movement from Mouth**	Seconds	1.32 (0.43)	1.31 (0.53)	−0.01	[−0.54, 0.53]
**Duration of Movement (total)**	Seconds	3.46 (0.91)	3.51 (1.08)	−0.07	[−0.63, 0.48]
**Range of Pitch**	Degrees	43.74 (21.79)	45.60 (18.58)	−0.27	[−23.47, 22.93]
**Range of Roll**	Degrees	54.95 (28.50)	56.81 (27.37)	−1.32	[−27.16, 24.51]
**Peak Velocity to Mouth**	m/s	0.42 (0.19)	0.23 (0.13)	0.18	[−0.20, 0.56]
**Peak Velocity Down**	m/s	0.49 (0.31)	0.19 (0.09)	0.31	[−0.07, 0.68]
**Fluency - Acceleration Zero Crossing (total)**	Number	4.67 (6.54)	3.00 (3.33)	1.8	[−7.23, 10.82]

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
