# Peer review of "DataSpoon: Validation of an Instrumented Spoon for Assessment of Self-Feeding"

_sensors, 2020, doi:10.3390/s20072114_

Round 1
Reviewer 1 Report
In this study the authors provides an experiment whose goal is to validate an instrumented fork for the assessment of self-feeding in children with motor deficits. In general I found the setup, method and rational driving data analysis clearly exposed in the manuscript. I found no obvious typos, which suggest careful examination before submission. I have only one concern, albeit major. The title refers to the validation of a spoon intended “for children with motor impairments”. However the data presented in the current manuscript refer only to healthy adults. As acknowledged briefly in the conclusion, the kinematics of hand movements in children with cerebral palsy is expected to be different, presumably jerkier, or conversely less fluid/smooth as compared to the healthy adults tested here. As a result if the true goal of this study was to validate an instrumented spoon that is intended later for children with motor disabilities, one may wonder why the authors did not test their spoon with disabled children. Note that this issue is particularly relevant since the reliability for tangential velocity and acceleration is lower than for any the other measures investigated here. Overall I think either the title should be restricted to the population that was tested here (healthy adults), either some extra data collected from disabled kids should be provided to reliably validate the spoon for this atypical population.

Reviewer 2 Report
General comment:
The paper describe a validation study of a spoon with an integrated motion sensor. Even though the study has potential, some revisions are needed prior to publication. For example, the method section does not describe all parts of the data analysis in a clear way, and the discussion part is very limited. Also, the conclusion should be immersed.
Specific comments:
Methods:
Figure 2 – this figure should be improved. Pitch and jaw directions are unclear from the figure.
Row 76-82: Describe how did you handle that different sample frequencies were used? Did you downsample the TrakStar data?
Row 105- Describe in the method part how the synchronization were done. You mention this in the figure caption in Figure 3 but this should also be describes in the method section. Pleas describe more clear how the pitch motion was detected (automatically or by manual inspection?)
Row 115-125: Why did you use the acceleration signal to derive velocity instead of using the gyroscopic angular velocity? The integration step may add a drift component that contribute to lower ICC’s seen in Table 1.
Page 4- row 119 – please explain how you compensated for the g vector. It is not clear. If you only subtract the baseline, this will induce errors as the spoon starts to rotate.
Row 139-141 – remove this text.
Results:
Table 1: Make sure it is easy to identify which values are ICC values and which values that are p values. For example, state in the table caption that 95% CI are given within brackets, and so on.
Discussion:
The discussion part is very sparse. For example, you use three different types of grip, but you don’t discuss whether the system enables a differentiation between these types of grips, something that is of high interest if you want to use the spoon to classify hand function. Please discuss your results regarding
- Which of your measures that can be used to discriminate between different types of grips? Are these certain measures reliable according to your study?
- Also discuss why faster movements are less reliable than slow movements according to your study
Row 174-176: Rephrase this sentence: 2.2% and 64.5% of which set of papers from which years?
Row 167-169: Rephrase this sentence. The tangential velocity are unreliable based on your study (not only “lower”). This should be clearer.
Row 177 – once again, why focus on linear velocity instead of angular velocity? Then you do not get the integration error from integrating the acceleration signal.
Conclusions:
State more clear which measures you recommend when describing self-feeding kinematics and which measures that are unreliable based on your study.
Reviewer 3 Report
In this paper the authors focus on the ability of an instrumented spoon to assess self-feeding kinematics. The low-cost inertial sensor which is part of the DataSpoon is compared with gold standard sensor. The device has very promising applications and checking its sensor’s accuracy is a necessary step in view of further developments.
The paper is clearly written and the results deserve publication ; however, in its current form, it lacks information concerning data analysis and “real clinical” applications of the DataSpoon. Here are the comments the authors should take into account.
L40 : The first sentence is not clear enough. Do the authors mean “Sensor-based assessment of movement kinematics is currently…” ?
L52 : motion sensors could also be used to give a feedback (e.g. auditory) like in the case of a kettle
https://www.sciencedirect.com/science/article/abs/pii/S0003687015000617
Can the authors elaborate on this? Maybe it should be included in the discussion : would the DataSpoon help the user in real time his/her self-feeding tasks ?
L53 : [DataSpoon; 12,13] à [12,13] ?
L55 : At this stage of the introduction it could be interesting to give explicit examples : which kind of kinematical data is helpful for clinicians and why ?
L80 : Please give the price of trakSTAR in the text in order to compare with the inertial sensor. Please also add a table comparing both sensors : size, mass, accuracy, range, etc.
L98 : Why is the rotated power grip testing the limits of detection of movement landmarks ? The authors should give more details. Please also link Figs 1B,C,D to children with CP : Are they generally able to take a spoon as in Fig 1B or are nonstandard ways expected ?
L106 : The inertial sensor’s gyroscope give angular speeds. Please give more details on your signal processing : what is the method chosen to integrate angular speed and get angles ? How do you remove drift ?
L113 : I find Pitch and Yaw difficult to differentiate from Fig 2. Could the author add some information to clarify the figure ?
L124 : The number of zero crossings in the acceleration profile is called “fluency”. Is there an intuitive way of linking the zero crossings and the concept of fluency ?
L125 : Fig 3 is important to understand the results. However only the angles vs time are displayed. A figure showing the velocity and acceleration profiles should be present too and show the reader how “peak velocity to mouth”, “peak velocity down” and “fluency” are computed.
L126 : Please add a graphical representation of the range.
L139-141 : I think this paragraph is part of the Sensor’s template and should be removed.
L145 : Why are some events impossible to be identified ? Please give details. Could it be a problem in “real-life” situations ?
L148 : Table 1 would be clearer if all the p-values appeared at the same line.
L158 : Please define NA in the caption of Table 2.
L165 : Why would the DataSpoon be restricted to children with CP ? From the present paper it seems that the device may study self-feeding kinematics of any patient. The authors should either list all the possible applications of DataSpoon, or give explicit reasons of its use in children with CP. In particular, what features of movements may be clinically important ? Can the authors give example of features that clinicians need ?
L176 : Some instrumented spoons exist to allow self-feeding of patients with tremor, see e.g. https://ajot.aota.org/article.aspx?articleid=2728507
Would a system such as the Liftware be relevant in children with CP ? Please compare with DataSpoon.
Round 2
Reviewer 1 Report
The authors have decided not to collect extra data regarding children but have consented to change their title accordingly. Still in the response letter they claim that the feasibility of data spoon has been already demonstrated in children with CP [26]. If correct, I am puzzled about the (true) motivation for carrying an extra study in adults only, and why this earlier work is not quoted upfront in the introduction (rather than in the discussion). Providing a clearer rational for this extra study (in adults only) would help claryfing the impact/utility of this follow up study.
Reviewer 3 Report
The authors have carefully addressed all my comments and modified the text accordingly. Therefore I think the paper ready for publication.
Author Response
We thank the reviewer for their thoughtful comments.